# Learning Generalizable Dexterous Manipulation from Human Grasp Affordance

**Yueh-Hua Wu**[*] **Jiashun Wang**[*] **Xiaolong Wang**

UC San Diego

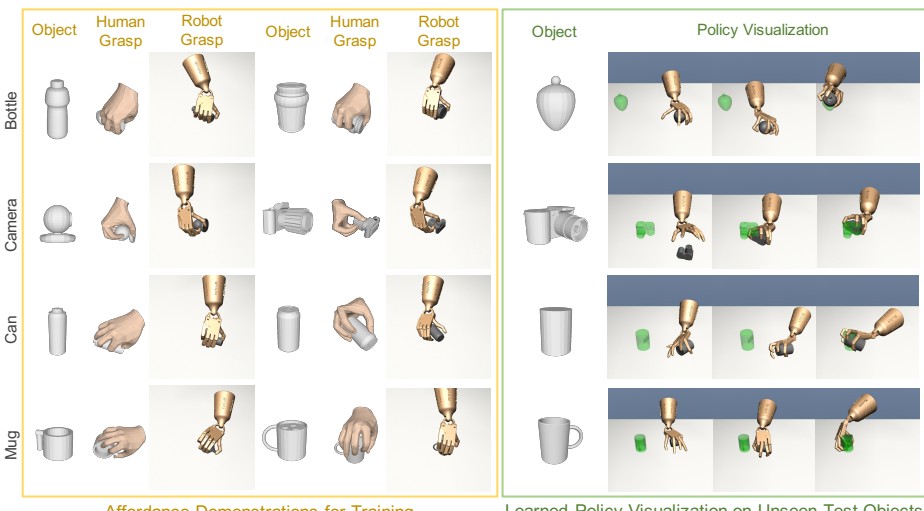

Figure 1: Examples of our affordance demonstrations and learned policies. **Left**: We visualize two groups of demonstrations for each object category. In each group, we visualize the given instance, generated human grasp, and the robot grasp for demonstrations. **Right**: We train a policy using imitation learning with the demonstrations, and visualize the learned policy relocating the unseen object.

**Abstract:** Dexterous manipulation with a multi-finger hand is one of the most challenging problems in robotics. While recent progress in imitation learning has largely improved the sample efficiency compared to Reinforcement Learning, the learned policy can hardly generalize to manipulate novel objects, given limited expert demonstrations. In this paper, we propose to learn dexterous manipulation using large-scale demonstrations with diverse 3D objects in a category, which are generated from a human grasp affordance model. This generalizes the policy to novel object instances within the same category. To train the policy, we propose a novel imitation learning objective jointly with a geometric representation learning objective using our demonstrations. By experimenting with relocating diverse objects in simulation, we show that our approach outperforms baselines with a large margin when manipulating novel objects. We also ablate the importance of 3D object representation learning for manipulation. We include videos and code on the project website - `https://kristery.github.io/ILAD/`.

**Keywords:** Dexterous manipulation; Affordance modeling; Imitation learning

## 1 Introduction

Human hands provide the primary means for our daily life interactions with the physical world. Our hands exhibit tremendous flexibility in operating objects around us. To enable the robot the same flexibility in assisting humans in daily life, dexterous manipulation with multi-finger robot hands

---

[*]indicates equal contributions.

6th Conference on Robot Learning (CoRL 2022), Auckland, New Zealand.

has been one of the core problems in robotics. At the same time, it is one of the most challenging problems in robotics given its high Degree-of-Freedom joints (e.g., 24 to 30 DoF). While recent progress in Reinforcement Learning (RL) has shown encouraging results on complex dexterous manipulation [1, 2, 3], it is still limited by the requirement of a large number of samples in training, and the trained policy can hardly generalize to novel objects during deployment.

To improve the sample efficiency in training, one promising direction is to perform imitation learning from human demonstrations [4, 5, 6, 7, 8, 9]. The expert demonstrations for dexterous manipulation can be collected by a human from teleoperation in a Virtual Reality (VR) [5], using Mocap [10] and videos [8]. Guided by human demonstrations, it not only reduces sample complexity in learning but also helps robot hands perform human-like and safe behaviors. However, the current setup on data collection largely limits the scale and diversity of the demonstrations. For instance, data collection with VR in [5] only leads to 25 demonstrations per task with one single object instance. With limited data, the learned policy can hardly generalize and transfer to unseen objects in test time. Instead of focusing on small scale of data, can we obtain larger scale demonstrations from reasoning the key interactions between hand and objects?

In this paper, we propose to leverage the human grasp affordance model for generalizing dexterous manipulation to novel object instances in the same category. We will first generate large-scale demonstrations on human hands interacting with diverse objects within the same category from affordance reasoning (left columns in Fig. 1). Specifically, we can generate a hand grasp pose and a way to contact by leveraging the human grasp affordance model [11]. We utilize motion planning to generate a trajectory that moves the robot hand from a start state to the target grasp. This trajectory provides a partial demonstration of how the robot hand can reach and stably grasp the object like humans do, preparing for the downstream tasks. We then use imitation learning to train a policy by augmenting RL with these demonstrations and test on unseen objects (right columns in Fig. 1). Our policy takes the object point cloud and the robot hand state as inputs for decision making. We tackle generalization by jointly learning: (i) skill generalization with a new imitation learning objective; and (ii) geometric representation generalization with a behavior cloning objective. We illustrate these two components of our learning approach as following.

*Imitation Learning Objective.* To learn the policy, we augment RL with our generated demonstrations for imitation learning. Previous approaches weighted all demonstrations equally during learning [5, 7]. In order to take advantage of diverse and large-scale demonstrations, we propose a novel ranking function to encourage the policy to learn from trajectories that it is less likely to reproduce. In addition, we estimate advantage values for state-action pairs from demonstrations with a growing weight so that the policy can still benefit from the given demonstrations at late training phases.

*Geometric Representation Learning.* The policy needs to understand the object shape given the point cloud inputs to manipulate it accordingly. We utilize PointNet [12] to encode the input object and pre-train the representation with a behavior cloning task using our large-scale demonstrations. As the policy interacts with the environment during learning, we collect new data to continue fine-tuning the PointNet with behavior cloning. Our pipeline jointly optimizes the imitation learning objective for skill generalization and the behavior cloning objective for representation generalization.

We perform experiments in simulation with five different object categories. We train one policy for each object category on the *relocate* task, which requires the multi-finger robot hand to relocate an object instance from an initial position to a goal location. We focus on evaluating the metric on generalization to relocate novel objects that are not seen in training. We not only observe significant improvement over RL and imitation learning baselines but also ablate the effectiveness of our novel imitation learning objective and geometric representation learning using our demonstrations.

We highlight our main contributions as following: (i) A novel approach on generating large-scale dexterous manipulation demonstrations on diverse objects; (ii) A novel imitation learning objective and 3D geometric representation learning approach for generalizing dexterous manipulation; (iii) Significant improvement over baselines for dexterous manipulation on novel objects.

## 2 Related Work

**Dexterous Manipulation.** Dexterous manipulation with a multi-finger hand has been one of the core robotics problems [13, 14, 15, 16, 17, 18, 19, 4, 20, 21, 22, 23, 24]. Recent success has been shown in using Reinforcement Learning (RL) for solving complex dexterous manipulation tasks [1, 2]. Extending from this line of research, recent efforts [25, 26] have been made on using model-free RL to generalize diverse object in-hand reorientation. While these works have shown encouraging results, it is unclear how well it works for grasping and relocating objects, which is the main focus for our work. We emphasize that *both types of manipulation tasks are important problems that have wide applications, but with different focuses*. Sample complexity and unrealistic actions are still challenges for these approaches. We will compare to this line of approaches in our experiments. Our work is more related to recent efforts on using affordance and contact reasoning to design an auxiliary loss to guide human-like dexterous grasping [27]. However, the small scale of contact examples limits generalization ability of the policy, and the policy is not required to handle different environment configurations with different goals. In this paper, we propose a novel method to generate large-scale demonstrations from grasp affordance, and novel imitation learning algorithms for better generalization on a more challenging task setting.

**Imitation learning.** Imitation learning aims at recovering the expert policy that generates the given demonstrations. It includes using behavior cloning [28, 29, 30], Inverse Reinforcement Learning (IRL) [31, 32, 33, 34, 35, 36, 37, 38], and augmenting expert demonstrations to the online collected data for Reinforcement Learning [39, 40, 41, 5, 7]. Our work falls into the last line of research. For example, Rajeswaran et al. [5] propose to take maximum likelihood with demonstrations as an auxiliary term during RL training. However, they utilize perfect demonstrations collected via VR at a small scale. On the other hand, we propose to utilize imperfect partial demonstrations at a large scale for better generalization. Our work is also related to imitation learning and RL with videos [6, 42, 43, 44, 45, 46]. However, most of these works focus on relatively simple manipulators (i.e., a 2-jaw parallel gripper). Recent concurrent works have also proposed to collect dexterous manipulation demonstrations from human videos or visual teleoperation [8, 9, 47, 48], however, the scale of collected demonstrations is still much less than what we achieved in this paper. 120

**Visual representation learning for decision making.** Visual inputs provides not only the appearance but also the geometric information of the scene and object. A lot of recent efforts have been made on learning decision making directly from visual inputs [49, 50, 51, 52, 53, 54, 55, 56]. For example, Mu et al. [56] propose to train policy directly taking point clouds as inputs for manipulation tasks using a robot arm. Inspired by these works, our policy takes point clouds as inputs and uses a PointNet [12] to capture the object shape for manipulation. To improve generalization, we leverage the affordance model on reasoning how humans grasp an object [57, 58, 59, 60, 11, 61, 62] to generate large-scale demonstrations for training the PointNet representation.

## 3 Method

We propose to learn a policy using imitation learning with demonstrations generated from the human grasp affordance model. Our policy takes the object point clouds together with the hand joint states as inputs for decision making. We introduce our approach **I**mitation **L**earning from **A**ffordance **D**emonstration (ILAD) as a 2-stage pipeline:

(i) **Affordance Demonstration Generation**. We leverage a state-of-the-art affordance model GraspCVAE [11] to generate diverse grasps on diverse objects within the same category. With the generated grasps, we utilize motion planning to obtain trajectories to reach these grasps. While these trajectories do not show how to perform a particular task, they can serve as partial demonstrations for guiding our policy to achieve the right contacts in grasping.

(ii) **Imitation Learning with Representation Learning**. We propose a novel imitation learning objective to learn the policy with the affordance demonstrations. As we utilize a PointNet [12] encoder to extract the 3D object shape information, we propose a 3D geometric representation learning approach jointly with imitation learning.

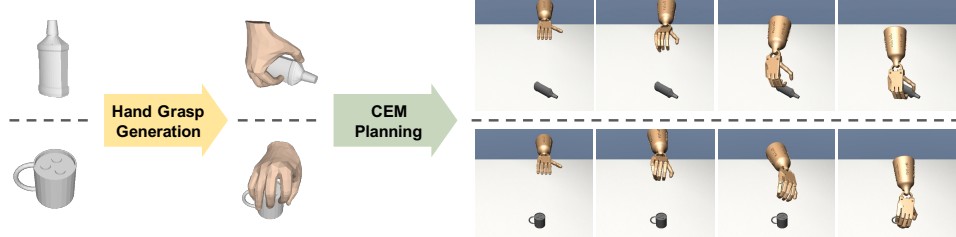

Figure 2: Affordance demonstration generation. We first generate the hand grasp on a given object and then use CEM for planning to reach the target grasp position. We provide two examples of this pipeline.

## 3.1 Affordance Demonstration Generation

We propose to generate demonstrations from human grasp affordance in 2 steps as shown in Fig. 2.

**Grasp generation.** Given diverse 3D objects, we adopt the pre-trained GraspCVAE model [11] to generate diverse human grasps for each object. Specifically, the GraspCVAE will take the object point cloud observation as inputs, and outputs the grasp pose represented by the MANO [63] model parameterized by shape parameter $\beta$ and pose parameter $\alpha$. With these parameters, we can compute the human hand joints using the forward kinematics function $j^h = \mathcal{J}_h(\beta, \alpha)$.

**Motion planning for grasp trajectory.** Given the target grasp hand joints $j^h$, our goal is to find an robot hand action sequence $a_1, ..., a_K$ which generates a robot hand joint sequence $j_0^r, ..., j_K^r$ so that the last robot hand joint positions $j_K^r$ reaches $j^h$. Note the initial robot hand joints $j_0^r$ are given. The objective for motion planning is $\min_{a_1,...,a_K} \|j_K^r - j^h\|^2 + \lambda\|p_K - p_1\|^2$, where $p_1$ and $p_K$ are object poses at time step 1 and $K$, and constant $\lambda = 10$. The first term of the objective encourages the robot hand to reach the human grasp, and the second term prevents the object from moving during the process. We use cross-entropy method (CEM) [64] for motion planning given this objective. We first sample actions from the Gaussian distribution $\mathcal{G}(\mu, diag(\sigma^2))$ and pick a fixed number of candidates with lower costs. We update the $\mu$ and $diag(\sigma^2)$ with these candidates and repeat iteratively and use model predictive control (MPC) to execute the planned trajectories until the objective below a threshold $\delta$.

In the following subsections, we will first introduce our novel imitation learning objective using these demonstrations, and then the policy training pipeline using both the imitation objective and a geometric representation learning objective to enhance the generalizability of the learned policy.

## 3.2 Imitation Learning Objective

We perform imitation learning using demonstrations generated from our planning algorithm, under a setting where a reward function for Reinforcement Learning and demonstrations are both given.

**Preliminaries.** We first consider a standard Markov Decision Process (MDP). It is represented by a tuple $\langle S, A, P, R, \gamma \rangle$, where $S$ and $A$ are state and action space, $P(s_{t+1}|s_t, a_t)$ is the transition density of state $s_{t+1}$ at step $t + 1$ given action $a_t$ made under state $s_t$, $R(s, a)$ is the reward, and $\gamma$ is the discount factor. The goal of RL is to maximize the expected reward with a policy $\pi(a|s)$.

We build our approach upon an imitation learning baseline algorithm called Demo Augmented Policy Gradient (DAPG) [5]. The learning objective function at epoch $k$ can be represented as,

$$g_{aug} = \sum_{(s,a)\in D_{\pi_\theta}} \nabla_\theta \ln \pi_\theta(a|s) A^{\pi_\theta}(s,a) + \sum_{(s,a)\in D_{\pi_E}} \nabla_\theta \ln \pi_\theta(a|s) \lambda_0 \lambda_1^k \max_{(s,a)\in D_{\pi_\theta}} A^{\pi_\theta}(s,a),$$

where $A^{\pi_\theta}$ is the advantage function [65] that is used to estimate the difference of the discounted reward sum starting from $(s, a)$ and $s$ according to policy $\pi_\theta$, $D_{\pi_E}$ are state-action pairs from the expert demonstrations, $D_{\pi_{\pi_\theta}}$ are state-action pairs collected with policy $\pi_\theta$, and $\lambda_0$ and $\lambda_1$ hyperparameters. In the implementation of [5], $\max_{(s,a)\in D_{\pi_\theta}} A^{\pi_\theta}(s,a)$ is set to 1 for stability.

**Learning from partial demonstrations with multiple objects.** For generalizing dexterous manipulation to multiple objects, where there are easier and more challenging shapes, the demonstrations

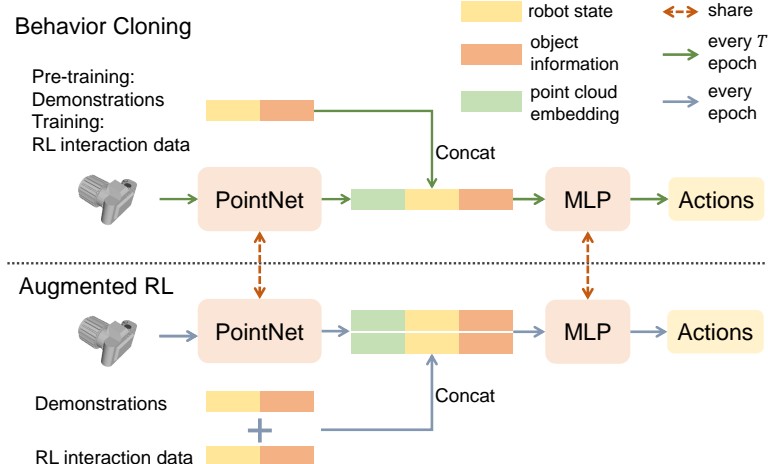

Figure 3: Training pipeline for the proposed behavior cloning and augmented reinforcement learning (augmented RL). For training, every $T$ epoch, we finetune the PointNet $\theta_{pc}$ using the objective (Eq. 2) with respect to $\theta_{pc}$ with the RL interaction data. During imitation learning, we update the MLP $\theta_p$ by estimating the gradient (Eq. 1) with the given demonstrations and the RL interaction data.

should not be taken equally during training. We propose to adaptively rank the demonstrations based on the difficulties of objects and the learning progress of the policy with the objective,

$$
g_{ILAD} = \sum_{(s,a) \in D_{\pi_\theta}} \nabla_\theta \ln \pi_\theta(a|s) A^{\pi_\theta}(s,a) + \sum_{(s,a) \in D_{\pi_E}} \nabla_\theta \ln \pi_\theta(a|s) \lambda_0 \lambda_1^k w_k(s,a) +
$$
$$
\sum_{(s,a) \in D_{\pi_E}} \nabla_\theta \ln \pi_\theta(a|s) \lambda_0'(1 - \lambda_1^k) A_\phi^{\pi_\theta}(s,a), \tag{1}
$$

where $w_k(s,a)$ in the second term is computed as the negative of a normalized log likelihood to encourage the policy to learn from trajectories that it is hard to reproduce by the current policy. $A_\phi^{\pi_\theta}(s,a)$ in the third term is an advantage function estimated with a model parameterized by $\phi$ for state-action pairs of the demonstrations. Formally, $w_k(s,a)$ can be represented as a normalized value (scaled between 0 and 1) of the negative of the log likelihood $w_k(s,a) = \frac{l_k(\tau_{s,a}) - \min_\tau l_k(\tau)}{\max_\tau l_k(\tau) - \min_\tau l_k(\tau)}$, where $\tau_{s,a}$ is a trajectory from the demonstrations that contains state-action pair $(s,a)$, and the negative of the log likelihood for a trajectory $l_k(\tau) = -\frac{1}{|\tau|} \sum_{(s,a) \in \tau} \log \Pr(s,a|\pi_\theta)$.

We explain our key innovations on the second term and the third term in Eq. 1 as following.

**Normalized likelihood weights** in the second term in Eq. 1. In our experiments, we find that the policy could easily learn to manipulate a certain kind of object while neglecting the others. To encourage the policy to generalize on diverse objects, we dynamically assign larger weights $w_k(s,a)$ for demonstrations that have a smaller likelihood in the current epoch.

**Advantage approximation for demonstrations** in the third term in Eq. 1 is designed to further elevate the utilization of demonstrations. While in the previous approach [5] the advantage function for demonstrations is taken as 1 as mentioned, we approximate the true advantage for a more accurate estimation of the gradients. We train a neural network $A_\phi^{\pi_\theta}(s,a)$ to predict the advantage function. This new advantage function is trained with data collected by RL and applied to the partial demonstrations. We set $\lambda_0' = 0.1\lambda_0$ across all experiments to prevent excessive parameter tuning.

### 3.3 Policy Training with Geometric Representation

Our policy takes both the point cloud of the object, object 6D pose, robot hand joint states as inputs, and predicts the actions. Specifically, to represent the object shape, we utilize the PointNet [25] encoder $\theta_{pc}$ for the point cloud inputs. Given the point cloud embedding, we concatenate it with the object 6D pose parameters and hand joint states together and forward them together to a 3-layer MLP $\theta_p$ network for decision making. Thus the policy network is parameterized by $\theta = \{\theta_{pc}, \theta_p\}$.

To perform training, besides optimizing towards the imitation learning objective, we design a geometric representation learning objective for training the PointNet jointly. Our overall model architecture and training pipeline are visualized in Fig. 3.

**Behavior cloning for geometric representation learning.** We utilize behavior cloning to provide an objective to train our PointNet encoder. We obtain the training data directly from the examples $D_{\pi_\theta}$ collected during the interaction with the environment in policy learning. Specifically, the behavior cloning objective can be represented as,

$$\mathcal{L}_{bc} = \frac{1}{|D_{\pi_\theta}|} \sum_{(s,a) \in D_{\pi_\theta}} \|\pi_\theta(s) - a\|^2, \tag{2}$$

where we still utilize the whole network $\theta = \{\theta_{pc}, \theta_p\}$ including the decision making MLP $\theta_p$ to compute the loss, we only optimize the PointNet parameters $\theta_{pc}$ through backpropagation. This part of training corresponds to the upper part of Fig. 3.

**Pre-training.** The same objective Eq. 2 can also be used to pre-train the PointNet encoder and policy network before policy learning with demonstrations $D_{\pi_E}$.

**Joint learning with both objectives.** We train our policy jointly with both the imitation learning objective and the behavior cloning objective as shown in Fig. 3. Empirically, we find training the PointNet with policy gradient in RL (Eq. 1) makes the representation unstable for decision making. The variance of gradients is much larger in RL and very challenging to learn an encoder with high-dimensional inputs directly [51, 52]. Thus, we propose to share the network parameters $\theta = \{\theta_{pc}, \theta_p\}$ for both objectives, but use policy gradient to optimize the decision making MLP $\theta_p$ and behavior cloning to optimize the PointNet encoder $\theta_{pc}$. That is, the gradients of Eq. 1 becomes with respect to $\theta_p$, instead of $\theta$.

---

**Algorithm 1** ILAD Pre-Training and Joint Learning

---
1: **Input:** partial demonstrations $D_{\pi_E}$, PointNet $\theta_{pc}$, policy network $\theta_p$
2: Pre-train $\theta_{pc}$ and $\theta_p$, according to Eq. 2
3: **for** $t = 0, 1, 2, ...$ **do**
4:     Sample trajectories $D_{\pi_\theta} = \{(s_i, a_i)\}_{i=1}^{n}$
5:     **if** $t \equiv 0 \pmod{T}$ **then**
6:         Update $\theta_{pc}$, according to Eq. 2
7:     **end if**
8:     Update $\theta_p$, according to Eq. 1
9: **end for**

---

For better stability, we perform slower updates on the PointNet encoder so that the decision-making learns from similar representations over time. Specifically, while we update the MLP $\theta_p$ for every epoch using policy gradients, we perform an update on the PointNet encoder $\theta_{pc}$ every $T$ epochs using behavior cloning. In Fig. 3, arrows with different colors represent different update strategies. We summarize our learning procedure in Algorithm 1.

## 4 Experiments

### 4.1 Experiment and Comparison Settings.

We conduct experiments on *Relocate* task with five categories: bottle, remote, mug, can, and camera. In this task, an object is placed on a table with random orientation and position and the robot is required to grasp the object and move it to a random target position. For each category, we use 40 objects for training and about 30 unseen objects (differ by category) for testing to evaluate the generalizability. The unseen objects did not appear during training but are within the same category as training objects. Both training and testing objects are from ShapeNet [66]. There are two settings of the demonstration size: 100 and 1000 for each category.

**Baselines and Compared Methods.** We adopt TRPO [67] as our RL baseline. We also perform comparison to DAPG [5]. For fair comparison, we use the same PointNet pre-training method as ILAD and use it to process point clouds inputs for DAPG, namely **DAPG+PC**. Since the code of [26] is not released and it is using a different simulator, we construct a baseline using RL with pointcloud inputs as an approximation. We add the joint learning of PointNet for this baseline, namely **RL+PC** where PC represents "PointCloud". Note the demonstrations are not adopted in this baseline. We use

| Model | Bottle | Remote | Mug | Can | Camera | **Average** |
|---|---|---|---|---|---|---|
| RL | $0.00 \pm 0.00$ | $0.62 \pm 0.24$ | $0.01 \pm 0.01$ | $0.00 \pm 0.00$ | $0.15 \pm 0.20$ | $0.16 \pm 0.14$ |
| RL+PC | $0.00 \pm 0.00$ | $0.05 \pm 0.02$ | $0.37 \pm 0.11$ | $0.00 \pm 0.00$ | $0.09 \pm 0.04$ | $0.11 \pm 0.05$ |
| DAPG+PC | $0.52 \pm 0.11$ | $0.56 \pm 0.14$ | $0.73 \pm 0.16$ | $0.58 \pm 0.22$ | $0.66 \pm 0.12$ | $0.61 \pm 0.20$ |
| DAPG+PC (large) | $0.33 \pm 0.41$ | $0.79 \pm 0.02$ | $0.95 \pm 0.02$ | $0.62 \pm 0.21$ | $0.52 \pm 0.03$ | $0.64 \pm 0.21$ |
| ILAD ($T$=50) | $0.96 \pm 0.03$ | $0.90 \pm 0.02$ | $0.96 \pm 0.02$ | $0.63 \pm 0.35$ | $\mathbf{0.99 \pm 0.01}$ | $0.89 \pm 0.25$ |
| ILAD ($T$ =20, large) | $0.83 \pm 0.03$ | $\mathbf{0.95 \pm 0.03}$ | $\mathbf{0.99 \pm 0.01}$ | $\mathbf{0.97 \pm 0.01}$ | $0.91 \pm 0.02$ | $0.93 \pm 0.02$ |
| ILAD ($T$ =50, large) | $\mathbf{0.99 \pm 0.01}$ | $0.94 \pm 0.02$ | $0.96 \pm 0.03$ | $0.93 \pm 0.02$ | $\mathbf{0.99 \pm 0.02}$ | $\mathbf{0.96 \pm 0.02}$ |

Table 1: The success rate of the evaluated methods on **unseen objects**. The unseen objects are not shown during training. For better clarity, we use "large" to represent demonstration size of 1000 trajectories during training and demonstration size of 100 trajectories for others. $T$ is the updating interval.

| Model | Bottle | Remote | Mug | Can | Camera | **Average** |
|---|---|---|---|---|---|---|
| DAPG+PC | $0.58 \pm 0.17$ | $0.54 \pm 0.20$ | $0.70 \pm 0.23$ | $0.58 \pm 0.24$ | $0.64 \pm 0.16$ | $0.61 \pm 0.20$ |
| ILAD (w/o JL; $\lambda_0' = 0$) | $0.61 \pm 0.31$ | $0.52 \pm 0.05$ | $0.72 \pm 0.36$ | $0.61 \pm 0.32$ | $0.69 \pm 0.23$ | $0.63 \pm 0.28$ |
| ILAD (w/o JL) | $0.65 \pm 0.24$ | $0.57 \pm 0.26$ | $0.76 \pm 0.26$ | $0.66 \pm 0.33$ | $0.70 \pm 0.25$ | $0.67 \pm 0.27$ |
| ILAD | $\mathbf{0.95 \pm 0.03}$ | $\mathbf{0.91 \pm 0.04}$ | $\mathbf{0.94 \pm 0.05}$ | $\mathbf{0.67 \pm 0.45}$ | $\mathbf{0.99 \pm 0.01}$ | $\mathbf{0.89 \pm 0.20}$ |

Table 2: The success rate of the ablative baselines on **unseen objects**. The unseen objects are not shown during training. The performance is evaluated via 100 trials for five seeds. For clarity, we use "JL" for joint learning in the table. In this comparison, we set updating interval $T = 50$ for the joint learning.

the same RL algorithm (TRPO) with the same hyper-parameters for all approaches. We use the same PointNet encoder architecture as our method for both DAPG+PC and RL+PC. The performance is evaluated with **five individual random seeds** and the seeds are the same across all comparisons.

## 4.2 Main Comparisons

**Success rate.** We perform comparisons using both small and large number of demonstrations. We use "(large)" to denote training with large scale number of demonstrations, and small number of demonstrations otherwise. The results are presented in terms of *success rate* on **unseen objects that are not shown during training** in Tab. 1. The performance is evaluated via 100 trials for five random seeds. A trial is counted as a success when the final object position is within 0.1 unit length to the target. Both the initial object position and target position are randomized.

Tab. 1 shows that ILAD outperforms baselines with a large margin. The RL baseline only achieves an average success rate of 16%. While ILAD achieves an average success rate of 96%, which is about 43% improvement over DAPG+PC. At the same time, a larger number of demonstrations always performs better for both DAPG+PC and ILAD. This indicates the **importance of our demonstration generation method** which enables automatic large-scale demonstrations.

Tab. 1 also shows that the RL+PC baseline (approximation of [26]) fails to benefit from using point cloud inputs, as it achieves similar or worse results compared to pure RL. We attribute it to the fact that the relocate task for diverse objects is hard to learn without demonstrations, and the low learning efficiency of RL is insufficient to collect informative data for training the 3D representation. It validates that the performance of representation learning relies on the quality of datasets [68, 69].

**Joint-learning updating interval $T$.** We use different learning intervals $T = 20$ and $T = 50$ with large number of demonstrations (Tab. 1). Our method is robust to the choice of $T$ as we observe there are only small differences between them. Learning with larger $T$ achieves slightly better results as the representations change more smoothly for more stable RL training.

## 4.3 Ablation Study

**Generalization.** We summarize the success rate on **unseen objects** under the same category in Tab. 2. We show ILAD achieves much better results than DAPG+PC and using our new imitation objectives with $\lambda_0'$ is essential. The joint learning significantly enhances the success rate for unseen objects from $67\%$ to $89\%$ on average. This suggests that updating the encoded point cloud from behavior cloning objective in Eq. 2 greatly **alleviates the overfitting on the training objects**.

We show the average return of all the methods trained with five different random seeds using a small number (100 per category) of demonstrations in Fig. 4. The x-axis is the update iterations during training and the average return of the y-axis is normalized to the same range for all five categories.

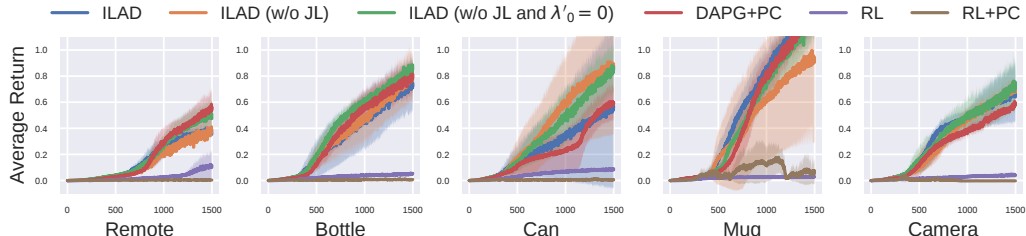

Figure 4: Learning curve of the proposed ILAD, its ablation, and the baselines. The return is estimated with **training objects**. For clarity, we use "JL" for joint learning. The x-axis is the update iterations and the average return of y-axis is normalized to the same range for all categories.

| Demonstration | Bottle |
|---|---|
| $\delta = 0.06$ w/o grasp poses | $0.01 \pm 0.01$ |
| $\delta = 0.1$ w/ grasp poses | $0.40 \pm 0.35$ |
| $\delta = 0.06$ w/ grasp poses | $\mathbf{0.65 \pm 0.24}$ |

Table 3: Demonstration quality ablation. The policies are evaluated on unseen bottle objects and they are trained with the same imitation learning approach.

| Demonstration | Rate |
|---|---|
| $\delta = 0.06$ w/o grasp poses | 0.29 |
| $\delta = 0.06$ w/ grasp poses | **0.71** |

Table 4: Demonstration quality user study. 71% users vote that ours are more natural and it shows our method can generate more human-like grasping demonstrations.

We emphasize Fig. 4 reports results on **training objects**, and both Tab. 1 and Tab. 2 report results on **unseen objects**. While all methods except for RL perform competitively in terms of average return from training environments in Fig. 4, fitting the same on training environments do not necessarily mean they can all transfer well in unseen objects. Among these methods, Tab. 1 and Tab. 2 both show that ILAD is able to achieve much better results on unseen objects compared to other approaches. **This comparison indicates the superior generalization power of our method on unseen objects**.

**Ablation on demonstration quality.** We ablate the performance of the policy on relocating bottles depending on different demonstration generation approaches in Tab. 3. We ablate threshold $\delta$ and approach with and without hand grasp information. For the approach without hand grasp pose, we minimize the distance between the robot palm hand and the object during planning. With the same threshold $\delta$, the demonstrations without hand grasp pose fail to provide sufficient information for imitation learning, indicating the **importance of modeling hand-object interactions**.

A smaller threshold $\delta$ indicates that the final states of the generated demonstration are closer to the desired grasp poses. Tab. 3 shows that the performance will drop without precise planning for grasps. This provides evidence that the generated demonstrations do not only facilitate the learning of the reaching phase but also give the agent a better instruction to grasp objects of different shapes. We also compare the proposed sampling method with Rapidly-exploring random trees (RRT) and we leave the results to supplementary materials due to the limited space.

**User study on naturalness.** We compare demonstrations generated with and without hand grasp from the affordance model and perform a user study on Amazon Mechanical Turk. We ask users to choose which demonstrations look more natural and the results are shown in Tab. 4. The survey suggests that 71% users would choose our demonstrations as more natural grasping demonstrations. Along with the ablation study in Tab. 3, it shows that naturalness is indeed a key factor to allow the policy to acquire a higher success rate on unseen objects.

## 5 Conclusion and limitation

**Conclusion.** We propose ILAD to generalize dexterous manipulation with point cloud inputs. To achieve generalization on diverse unseen objects, we introduce a novel approach to generate large-scale demonstrations and new imitation learning methods. We believe such ability of generalization is also essential for Sim2Real transfer for real robot dexterous hand manipulation. We are committed to **releasing our code, environments, and dataset**.

**Limitation.** While this paper has not include real robot experiments, we can perform randomization on object point clouds to simulate real-world occlusions, which **can potentially lead to direct Sim2Real deployment** of our method. The main focus of this work is unseen object generalization, and we will conduct the real-robot extension as our future work.

**Acknowledgments**

This work was supported, in part, by grants from NSF CCF-2112665 (TILOS), NSF 1730158 CI-New: Cognitive Hardware and Software Ecosystem Community Infrastructure (CHASE-CI), NSF ACI-1541349 CC*DNI Pacific Research Platform, the Industrial Technology Innovation Program (20018112, Development of autonomous manipulation and gripping technology using imitation learning based on visualtactile sensing) funded by the Ministry of Trade, Industry and Energy of the Republic of Korea, and gifts from Meta, Google, Qualcomm.

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
