# OpenReview forum: "Learning Generalizable Dexterous Manipulation from Human Grasp Affordance"
_robot-learning.org/CoRL/2022/Conference — CoRL 2022 Poster_

### Official Review · Reviewer_nXCy · 2022-07-29

**Originality:** Good
**Technical Quality:** Very Good
**Clarity Of Presentation:** Good
**Impact:** 3

**Recommendation:**

Weak Accept: I recommend accepting the paper, but will not argue for my recommendation if the majority of other reviewers have a different opinion.

**Summary:**

This paper mainly studies the problem of dexterous grasping. It first generates grasp demonstrations from a human grasp affordance model. Using the generated demonstrations it conduct demonstration-augmented RL to learn dexterous relocation. The proposed method also encode the point cloud observation with a PointNet and pre-train and finetune the PointNet with behavior cloning tasks.

**Issues:**

1. Include a baseline with the demonstration generation method and show the learned method can outperform it. Also analyze the reason behind.
2. Include a baseline without PointNet pre-training.

**Quality Of The Limitations Section:**

Limitations are addressed clearly

**Reviewer Expertise:**

3: The reviewer is fairly confident that the evaluation is correct

**Robotics Focus:**

Highly relevant to robotics but no hardware experiments

**Strengths And Weaknesses:**

Strengths:
1.  The proposed data generation pipeline (both grasping pose and approaching trajectories) is useful.
2. Pre-training and finetuning visual encoder with behavior cloning task is an interesting idea.
3. The performance of the proposed method is very impressive, surpassing the baselines with a significant margin.

Weakness:
1. The proposed task, relocation, consists of three phases: approaching, grasping, and lifting. The proposed method for demonstration generation seems to already handle the first two phases. For the last phase of lifting and placing it in a target pose, I think it can be implemented with some simple hard code. From the videos, I can see the proposed method mostly translates the object to the desired location and does not handle the orientation properly, so probably it's an even simpler task. The authors should include a baseline with the demonstration generation method combined with a hard-coded lifting policy and evaluate the result.
2. The benefit of pre-training is not demonstrated in the experiment results. The baseline with joint learning but without pre-training, the PointNet should be included. It would be stronger if the author could compare it with other point cloud network pre-training methods like PointContrast.

Minors:
1. Fig 3 is a little confusing:
  a. the arrangement of the legend of the figure looks like robot state <--> share, object information goes into every T epoch and point cloud embedding goes into every epoch. Leave more margin between the columns.
  b. it looks like the demonstration is the input to the MLP. Adding a line between demonstration and the output actions to show demonstration is also the supervision would make it clearer.
2. It would be great if the authors could try if the proposed method can generalize to objects of different categories other than the training ones.


**Summary Of Recommendation:**

I'm actually voting for borderline. The results in the simulation are pretty impressive, but I don’t think the argument stands on its ground if we can get the same result with the method that generates the demonstrations. If the authors can address my issues, I'm very likely to change my opinion.

-----

Post-rebuttal:

The author addressed my major issue and demonstrate the necessity of learning in the problem. So I change my recommendation from WR to WA.

---

> ### Author Response · Authors · 2022-08-26
> **Response to Reviewer nXCy**
>
> We thank the reviewer for the thoughtful comments. We address individual comments in the following.
>
> ---
>
> **Q**: “The proposed task, relocation, consists of three phases: approaching, grasping, and lifting. The proposed method for demonstration generation seems to already handle the first two phases. For the last phase of lifting and placing it in a target pose, I think it can be implemented with some simple hard code. From the videos, I can see the proposed method mostly translates the object to the desired location and does not handle the orientation properly, so probably it's an even simpler task. The authors should include a baseline with the demonstration generation method combined with a hard-coded lifting policy and evaluate the result.”
> ”
> **A**: We conduct an evaluation on our demonstration generation combined with a hard-coded lifting policy. Specifically, when the dexterous hand reaches the target grasp pose, we will tighten the joint position of the fingers, which will make the hand grasp the object as tightly as possible. After that, we give a hard-coded lift trajectory to the robot.  The comparison is shown as the table below. For each column, both approaches are trained with the corresponding categorical demonstrations.
>
> method | Bottle| Remote | Mug | Can | Camera
> --- | --- | ---| --- | --- | ---
> Hard Code| 0.10 $\pm$ 0.00 | 0.02 $\pm$ 0.00 | 0.16 $\pm$ 0.00 | 0.06 $\pm$ 0.00 | 0.15 $\pm$ 0.00
> ILAD | 0.99 $\pm$ 0.01 | 0.94 $\pm$ 0.02 | 0.96 $\pm$ 0.03 | 0.93 $\pm$ 0.02 | 0.99 $\pm$ 0.02
>
> We believe the difference is because the demonstration method is referring to a human-grasp pose and is not actually taking into account the physics for grasping. While it can provide a good guidance on the geometry level, it cannot directly work in the physical simulator. In our approach, we use this geometry / affordance guidance from the human demonstrations and combine it with RL to achieve better interactions with the physical world. The demonstrations could provide a more natural and generalizable grasp prior which is helpful for learning the policy.
>
> ---
>
> **Q**: “The benefit of pre-training is not demonstrated in the experiment results. The baseline with joint learning but without pre-training, the PointNet should be included.”
>
> **A**:  We summarize the ablation study on the pretraining in the table below. Our results suggest that the proposed method does benefit from the pretraining. Learning from high-dimensional data such as pointcloud and images may easily obtain a mediocre solution without a proper initialization. The pretraining part with the demonstration data provides the PointNet a rough distribution of the training objects and the reinforcement learning can learn a more robust policy.
>
> |       |Bottle      |
> |----------------|--------|
> |ILAD (w/o pretraining)|  $0.69\pm0.31$    |
> |ILAD  |$0.95\pm0.03$
>
> ---
>
> **Q**: “Fig 3 is a little confusing: a. the arrangement of the legend of the figure looks like robot state <--> share, object information goes into every T epoch and point cloud embedding goes into every epoch.”
>
> **A**: We thank the reviewer for providing a way to increase the clarity of Fig. 3. We will modify the figure to better indicate the source of state-action pairs.
>
> ---
>
> **Q**: “It would be great if the authors could try if the proposed method can generalize to objects of different categories other than the training ones.”
>
> **A**: We evaluate our category-level policy on objects from a **different** category. Specifically, we train our method and DAPG with Remote and then test on the other categories. We show the results in the following table. Our results indicate that our method can learn a more generalizable and robust policy. Even training with only the Remote instances, our method can successfully transfer the policy and achieve relatively high success rates on other categories. Meanwhile, DAPG failed to transfer the policy to other categories and achieve relatively low success rates.
>
> method | Mug | Can | Camera | Bottle| Remote
> --- | --- | --- | --- | --- | ---
> DAPG trained on remote | $0.38\pm0.31$ | $0.32\pm0.28$ | $0.28\pm0.23$ | $0.36\pm0.31$ | $0.54\pm0.20$
> ILAD trained on remote | $0.68\pm0.04$ | $0.77\pm0.04$ | $0.8\pm0.05$ | $0.77\pm0.02$ | $0.91\pm0.04$

---

> > ### Comment · Reviewer_nXCy · 2022-08-26
> > **Response to response**
> >
> > Thank you for your response! The results are good but some of the explanations are not very convincing.
> >
> > 1. Failure of the hard code policy:
> > Can you observe the rollouts and find when the hard code policy fails the most? Is it approaching the false position, not giving a very tight grasping, or lifting and reposing to a false pose? The current explanation is too high-level and abstract.
> >
> > 2. The benefit of pre-training:
> > Can you also provide a training curve comparison to better demonstrate how pre-training benefits performance?
> >
> > 3. Generalization to other categories:
> > The DAPG didn't get very good performance on the Remote category, so it's not very fair to say that it failed to transfer to other categories.

---

> > > ### Author Response · Authors · 2022-08-27
> > > **Thank you for the further comments**
> > >
> > > We thank the reviewer for more detailed questions. Following are our answers to the questions.
> > >
> > > **Q**: “Failure of the hard code policy: Can you observe the rollouts and find when the hard code policy fails the most? Is it approaching the false position, not giving a very tight grasping, or lifting and reposing to a false pose? The current explanation is too high-level and abstract.”
> > >
> > > **A**: According to our observation, the most failure scenarios appear because of not giving a very tight grasp. There are two main reasons for this. One reason is that the grasp synthesis based on the geometric generative model may not necessarily provide the foam closure specifically if we want to transfer from human hand to robot hand. Another reason is our current hard code is to tighten the fingers towards the palm, but because of the high degree of freedom of the dexterous hand, tightening towards the palm may not be the most ideal way. Thus using learning here is important. With our imitation learning pipeline, we can use human hands to provide priors and then use reinforcement learning to explore in order to find the suitable physical grasps for different objects.
> > >
> > > ---
> > >
> > >
> > > **Q**: “The benefit of pre-training: Can you also provide a training curve comparison to better demonstrate how pre-training benefits performance?”
> > >
> > > **A**: We show the [training curves here](https://drive.google.com/file/d/18n1Ev_rz_9sCxKcoyCLeTLaVfRUAxAwM/view?usp=sharing). The difference between ILAD w/ and w/o pretraining is significant even in the training phase.
> > >
> > > ---
> > >
> > >
> > > **Q**: “Generalization to other categories: The DAPG didn't get very good performance on the Remote category, so it's not very fair to say that it failed to transfer to other categories.”
> > >
> > > **A**: We additionally show the evaluation results training on the Bottle and the Mug category in the table below. The best success rate that DAPG obtained is on the Mug category but the success rate still falls from $0.70$ to as low as $0.26$ when evaluated on the other categories.
> > >
> > > method   | Mug | Can | Camera | Bottle | Remote
> > > ---|---|---|---|---|---
> > > DAPG trained on remote | $0.38\pm0.31$ | $0.32\pm0.28$ | $0.28\pm0.23$ | $0.36\pm0.31$ | $0.54\pm0.20$
> > > ILAD trained on remote | $0.68\pm0.04$ | $0.77\pm0.04$ | $0.8\pm0.05$ | $0.77\pm0.02$ | $0.91\pm0.04$
> > > DAPG trained on bottle | $0.50\pm0.34$ | $0.26\pm0.21$ | $0.34\pm0.31$ | $0.58\pm0.17$ | $0.32\pm0.16$
> > > ILAD trained on bottle | $0.82\pm0.11$ | $0.93\pm0.01$ | $0.87\pm0.04$ | $0.95\pm0.03$ | $0.88\pm0.04$
> > > DAPG trained on mug | $0.70\pm0.23$ | $0.31\pm0.28$ | $0.34\pm0.25$ | $0.37\pm0.33$ | $0.26\pm0.21$
> > > ILAD trained on mug | $0.94\pm0.05$ | $0.90\pm0.07$ | $0.88\pm0.02$ | $0.75\pm0.15$ | $0.68\pm0.17$

---

> > > > ### Comment · Reviewer_nXCy · 2022-08-27
> > > > **Response to response**
> > > >
> > > > Thank you, your response addressed my issues.

---

### Official Review · Reviewer_WeMh · 2022-07-30

**Originality:** Good
**Technical Quality:** Good
**Clarity Of Presentation:** Fair
**Impact:** 3

**Recommendation:**

Weak Accept: I recommend accepting the paper, but will not argue for my recommendation if the majority of other reviewers have a different opinion.

**Summary:**

This paper presents a pipeline for a robot hand to grasp onto different categories of objects and move the object to a target pose. The target grasp of the hand on the object is generated by an affordance model GraspCVAE, which was then reached using CEM for planning. The next step is to perform imitation learning based on DAPG using demonstrations generated from the previous planning algorithm. The policy encourages learning from trajectories that is hard to reproduce by dynamically assigning a normalized likelihood weight during training. A neural network is trained to predict the advantage function that is applied to the partial demonstration. PointNet is used to encode object shape, which is jointly trained with the imitation learning MLP.  The proposed method was compared with multiple baselines, and the ablation study shows the advantage of joint learning.

**Issues:**

* Figure 4: “JT” in the legends and “JL” in the caption.
* Line 155: $\\rho_{\\pi_{\\theta}}$ instead of $D_{\\pi_{\\theta}}$ in the equation above.
* The equation between line 151 and 152 is missing the number.
* Figure 3. Since “Behavior cloning” is a type of “Imitation learning”, the labeling can be confusing and makes the figure difficult to understand.
* The user study on naturalness demonstrated the benefit of affordance model (a prior work used in this pipeline) rather than that of the proposed method, which does not seem to be very useful in this paper.
* It’s a bit of stretch to use the phrase “dexterous manipulation” in the paper. The action generated by the hands are mostly two stages: (1) grasp onto the object, and (2) move the object to a target pose. The proposed policy is more or less learning a grasp pose of the hand. While the hand does manipulate the object to a final pose. During this stage there’s no relative motion between the object and the fingers, and such manipulation can be achieved through simple inverse kinematics.
* Please try to have a more elaborated limitation section rather than just focusing on explaining why there is no real-world experiment.


**Quality Of The Limitations Section:**

Limitations are not well addressed

**Reviewer Expertise:**

5: The reviewer is absolutely certain that the evaluation is correct and very familiar with the relevant literature

**Robotics Focus:**

Highly relevant to robotics but no hardware experiments

**Strengths And Weaknesses:**

**Strength:**
* Extensive experiments to demonstrate the effectiveness of the proposed method in simulation.

**Weaknesses:**
* No real world demonstration.
* Clarity or the quality of writing needs improvement. (see “Issues” section for details)
* The scope claimed is not sufficiently backed by what’s demonstrated in the paper. For example the word “manipulation” is not in any of the technical sections. (see “Issues” section for details)


**Summary Of Recommendation:**

The main contribution of the paper is to propose a method to generate actions for grasping object in certain grasp pose and manipulating the grasped object to a target pose. The generated method has a high success rate during the experiment and has demonstrated certain generalizability to objects within the same category. The reviewer believes this work provides a novel approach to object grasping (fetching) than dexterous manipulation as claimed in the title and introduction. Nevertheless, it does showcase algorithmic innovations and demonstrated certain level of success.

---

> ### Author Response · Authors · 2022-08-26
> **Response to Reviewer WeMh**
>
> We thank the reviewer for the thoughtful comments. We address individual comments in the following.
>
> ---
>
> **Q**: “No real world demonstration.”
>
> **A**: We agree with the limitation of no real robot experiments in this paper, and we have actually mentioned it in our limitation section (line293). However, compared to previous work, we emphasize our method on joint training a generalizable point cloud representations for manipulation is one important step leading to Sim2Real transfer. We argue that we have provided sufficient technical contributions including a new way to obtain demonstrations and a new learning method in this paper. It will take more than one paper to achieve the Sim2Real goal given such a challenging problem.
>
> ---
>
> **Q**: “Clarity or the quality of writing needs improvement. (see “Issues” section for details). Figure 4: “JT” in the legends and “JL” in the caption. Line 155:  $\rho_{\pi_{\theta}}$ instead of $D_{\pi_{\theta}}$ in the equation above.”
>
> **A**: We thank the reviewer for detailed comments. We will fix these typos.
>
> ---
>
> **Q**: “The scope claimed is not sufficiently backed by what’s demonstrated in the paper. For example the word “manipulation” is not in any of the technical sections. (see “Issues” section for details). It’s a bit of stretch to use the phrase “dexterous manipulation” in the paper. The action generated by the hands are mostly two stages: (1) grasp onto the object, and (2) move the object to a target pose. The proposed policy is more or less learning a grasp pose of the hand. While the hand does manipulate the object to a final pose. During this stage there’s no relative motion between the object and the fingers, and such manipulation can be achieved through simple inverse kinematics.”
>
> **A**: In order to show the effectiveness of our pipeline on the manipulation task, we propose an additional pour task in this rebuttal. The goal is to use the robot hand to pour the particles in the mug into a container. Evaluation is based on the percentage of particles inside the container. We use the same pipeline as the relocate task. We show a [visualization result of our method here](https://drive.google.com/file/d/1ctFod2JjZcCDmnQ2-d3d8E2mtZaS0DW4/view?usp=sharing) and also show the success rate in the table below. We will include more details and results on the pour task in our future version.
> method | DAPG | ILAD
> --- | --- | ---
> success rate | $0.83\pm0.11$ | $0.91\pm0.16$
>
> ---
>
> **Q**: “The equation between line 151 and 152 is missing the number.”
>
> **A**: Thank you for pointing it out. We will fix this.
>
> ---
>
> **Q**: “Figure 3. Since “Behavior cloning” is a type of “Imitation learning”, the labeling can be confusing and makes the figure difficult to understand.”
>
> **A**: Thank you for the opinion. We will change imitation learning to Augmented Reinforcement Learning.
>
> ---
>
> **Q**: “The user study on naturalness demonstrated the benefit of affordance model (a prior work used in this pipeline) rather than that of the proposed method, which does not seem to be very useful in this paper.”
>
> **A**: A simple yet effective demonstration generation method is also part of our pipeline. We want to show our demonstrations are more natural and human-like.
>
> ---
>
> **Q**: “Please try to have a more elaborated limitation section rather than just focusing on explaining why there is no real-world experiment.”
>
> **A**: Thank you for the suggestion. We will improve the limitation section. Another limitation of this work is it currently can perform well with the 5-finger Adroit Hand. In principle the same approach can be applied to robot hands with a different morphology, but it is unclear yet how well it will transfer.

---

### Official Review · Reviewer_p8H4 · 2022-07-31

**Originality:** Good
**Technical Quality:** Good
**Clarity Of Presentation:** Very Good
**Impact:** 3

**Recommendation:**

Weak Accept: I recommend accepting the paper, but will not argue for my recommendation if the majority of other reviewers have a different opinion.

**Summary:**

This paper focuses on the problem of grasping categories of objects with dexterous hands. The paper is built on top of DAPG. The main contribution of the paper is to add a new term to DAPG so that it focuses more on the data that it does not work well. Cause without that term, the policy learns to ignore the difficult objects and focus on the objects that can be easily manipulated. The architecture of the model is pointnet++ for processing the point cloud along with object pose and joint states. All the experiments are done in simulation and the generalization is only considered for different instances within the same category.


**Issues:**

See weaknesses above.

**Quality Of The Limitations Section:**

Limitations are addressed clearly

**Reviewer Expertise:**

5: The reviewer is absolutely certain that the evaluation is correct and very familiar with the relevant literature

**Robotics Focus:**

Highly relevant to robotics but no hardware experiments

**Strengths And Weaknesses:**

Strengths:
- The problem of learning from human demonstration for dexterous manipulation is important

Weaknesses:
- Very limited novelty: Apart from the second term of eq. (2), there is nothing novel about the method.
- There are no real world experiments.
- It seems like the policies are trained for each category separately. So different policies are needed for different categories.


**Summary Of Recommendation:**

Limited novelty and no real robot experiments are my biggest concern for this paper.

Post Rebuttal: I appreciate the authors for the new results on all the categories and held out test categories. I improved my rating.

---

> ### Author Response · Authors · 2022-08-26
> **Response to Reviewer p8H4**
>
> We thank the reviewer for the thoughtful comments. We address individual comments in the following.
>
> ---
>
> **Q**: “Very limited novelty: Apart from the second term of eq. (2), there is nothing novel about the method.”
>
> **A**: We would like to emphasize that besides the new ranking term, we have two important technical contributions including: (i) A novel approach to generate large-scale demonstrations on diverse objects based on grasp affordance. Note that before this paper, the scale of demonstrations is < 50 for dexterous manipulation and we scale them to 1000. (ii) A novel joint training method that learns both the 3D point cloud representation and decision making at the same time, the effectiveness of it is shown in Table 2 in our experiments. These contributions have also been summarized in line68-line71 in our paper.
>
> ---
>
>
> **Q**: “There are no real world experiments.”
>
> **A**: We agree with the limitation of no real robot experiments in this paper, and we have actually mentioned it in our limitation section (line293). However, compared to previous work, we emphasize our method on joint training a generalizable point cloud representations for manipulation is one important step leading to Sim2Real transfer. We argue that we have provided sufficient technical contributions including a new way to obtain demonstrations and a new learning method in this paper. It will take more than one paper to achieve the Sim2Real goal given such a challenging problem.
>
> ---
>
> **Q**: “It seems like the policies are trained for each category separately. So different policies are needed for different categories.”
>
> **A**: We conduct an experiment by training the proposed ILAD with demonstrations from all five categories and evaluating on objects that are not shown during training. The results are summarized in the first table below. It shows that the proposed ILAD is capable of learning to manipulate unseen objects from different categories at the same time.
>
> |             |Mug         |Can        |Camera   |Bottle   |Remote   |
> |-------------|---------|---------|---|---|---|
> |ILAD trained on all categories|$0.91\pm0.05$|$0.93\pm0.07$|$0.83\pm0.12$|$0.86\pm0.05$   |$0.82\pm0.08$   |
>
> Besides training a new policy, we also evaluate our category-level policy on objects from a **different** category. Specifically, we train our method and DAPG with Remote and then test on the other categories. We show the results in the following table. Our results indicate that our method can learn a more generalizable and robust policy. Even training with only remote instances, our method can successfully transfer the policy and achieve relatively high success rates on other categories. Meanwhile, DAPG failed to transfer the policy to other categories and achieve relatively low success rates.
>
> method | Mug | Can | Camera | Bottle| Remote
> --- | --- | --- | --- | --- | ---
> DAPG trained on remote | $0.38\pm0.31$ | $0.32\pm0.28$ | $0.28\pm0.23$ | $0.36\pm0.31$ | $0.54\pm0.20$
> ILAD trained on remote | $0.68\pm0.04$ | $0.77\pm0.04$ | $0.8\pm0.05$ | $0.77\pm0.02$ | $0.91\pm0.04$

---

### Meta-Review · Area_Chair_bNt7 · 2022-08-02

**Recommendation:** Accept (Poster)
**Confidence:** 4

**Metareview:**

Strengths:
+ Reviewers agree that the method presented here is effective at better using demonstrations for learning manipulation tasks compared to the main baseline, DAPG.
+ The method allows the use of significantly more expert demonstrations that previously possible
+ The simulated evaluation is extensive and shows better performance.

Weaknesses:
- The demonstration is somewhat narrow in scope, for a single class of manipulation tasks, and only for objects of the same category as seen in training. Note that this aspect was addressed in discussion where authors provided data on cross-category testing.
- There is no validation on real robots.
- The technical advance is effective, yet also somewhat limited (a ranking term in the use of demonstrations for DAPG). This aspect was also addressed in discussion where authors highlighted the joint training of representation and decision making.

Additional comment: The fundamental improvement compared to DAPG seems to be for generalization, rather than training performance. How do authors explain such a specific improvement? What about their algorithm leads to better generalization, despite similar training speed? The authors provided an answer in discussion attributing the change to avoidance of overfitting via what is essentially a form of regularization.

**Best Paper Nomination:**

No

---

> ### Author Response · Authors · 2022-08-26
> **Response to Meta Reviewer bNt7**
>
> We thank the reviewers for their feedback. We are glad to see all the reviewers agree with the effectiveness of the presented method over the baselines and the importance of our problem on generalizing dexterous manipulation.
>
> ---
>
> **Q**: “The demonstration is somewhat narrow in scope, for a single class of manipulation tasks, and only for objects of the same category as seen in training.”
>
> **A**: We conduct an experiment by training the proposed ILAD with demonstrations from all five categories and evaluating on objects that are not shown during training. The results are summarized in the first table below. It shows that the proposed ILAD is capable of learning to manipulate unseen objects from different categories at the same time.
>
> |             |Mug         |Can        |Camera   |Bottle   |Remote   |
> |-------------|---------|---------|---|---|---|
> |ILAD trained on all categories|$0.91\pm0.05$|$0.93\pm0.07$|$0.83\pm0.12$|$0.86\pm0.05$   |$0.82\pm0.08$   |
>
> Besides training a new policy, we also evaluate our category-level policy on objects from a **different** category. Specifically, we train our method and DAPG with Remote and then test on the other categories. We show the results in the following table. Our results indicate that our method can learn a more generalizable and robust policy. Even training with only remote instances, our method can successfully transfer the policy and achieve relatively high success rates on other categories. Meanwhile, DAPG failed to transfer the policy to other categories and achieve relatively low success rates.
>
> method | Mug | Can | Camera | Bottle| Remote
> --- | --- | --- | --- | --- | ---
> DAPG trained on remote | $0.38\pm0.31$ | $0.32\pm0.28$ | $0.28\pm0.23$ | $0.36\pm0.31$ | $0.54\pm0.20$
> ILAD trained on remote | $0.68\pm0.04$ | $0.77\pm0.04$ | $0.8\pm0.05$ | $0.77\pm0.02$ | $0.91\pm0.04$
>
> ---
>
> **Q**: “There is no validation on real robots.”
>
> **A**: We agree with the limitation of no real robot experiments in this paper, and we have actually mentioned it in our limitation section (line293). However, compared to previous work, we emphasize our method on joint training a generalizable point cloud representations for manipulation is one important step leading to Sim2Real transfer. We argue that we have provided sufficient technical contributions including a new way to obtain demonstrations and a new learning method in this paper. It will take more than one paper to achieve the Sim2Real goal given such a challenging problem.
>
> ---
>
> **Q**: “The technical advance is effective, yet also somewhat limited (a ranking term in the use of demonstrations for DAPG).”
>
> **A**: Thank you for acknowledging the effectiveness of our approach. We would like to emphasize that besides the new ranking term, we have two important technical contributions including: (i) A novel approach to generate large-scale demonstrations on diverse objects based on grasp affordance. Note that before this paper, the scale of demonstrations is < 50 for dexterous manipulation and we scale them to 1000. (ii) A novel joint training method that learns both the 3D point cloud representation and decision making at the same time, the effectiveness of it is shown in Table 2 in our experiments. These contributions have also been summarized in line68-line71 in our paper.
>
> ---
>
> **Q**: “The fundamental improvement compared to DAPG seems to be for generalization, rather than training performance. How do authors explain such a specific improvement? What about their algorithm leads to better generalization, despite similar training speed?”
>
> **A**: We attribute the improvement of generalizalizability to the regularization effect from the behavior cloning objective. Overfitting has been an issue in machine learning and a variety of methods and techniques such as regularization have been proposed to alleviate overfitting from different perspectives. Reinforcement learning, unlike supervised learning, optimizes reward sum over trajectories and relies heavily on interactions with environments to correct value estimation. The problem setting of reinforcement learning makes it hard to apply conventional methods dealing with overfitting since suppressing the learning capability of reinforcement learning without much care will greatly decrease the efficiency of learning from interactions and make optimization over trajectories even harder. In our work, we can see our behavior cloning objective as a regularization term to alleviate overfitting to the RL objective. Since the state-action distribution for behavior cloning is close to what the policy would output, the regularization is not too strong to hurt the training speed much and does not require much parameter-tunning.